# Study on the Fatigue Crack Initiation and Growth Behavior in Bismuth- and Lead-Based Free-Cutting Brasses

**DOI:** 10.3390/ma15217488

**Published:** 2022-10-25

**Authors:** Kenichi Masuda, Sotomi Ishihara, Noriyasu Oguma, Minoru Ishiguro, Yoshinori Sakamoto, Mami Iwasaki

**Affiliations:** 1Department of Mechanical Engineering, University of Toyama, Gofuku 3190, Toyama 930-8555, Japan; 2National Institute of Technology, Toyama College, Toyama 939-8630, Japan

**Keywords:** free-cutting brass, fatigue life, fatigue crack initiation, fatigue crack growth, short surface crack, modified linear fracture mechanics parameter

## Abstract

Several studies have been conducted on the fatigue behavior of copper and 7-3, and 6-4 brasses. However, there have been fewer studies on the fatigue behavior and fatigue crack growth (FCG) properties of free-cutting brass, primarily because emphasis has been placed on the development of lead-free free-cutting brass. In this study, fatigue experiments were performed in the atmosphere at room temperature using three types of free-cutting, two types of bismuth (Bi)-based (with different grain sizes), and lead (Pb)-based brasses. It was found that lead-free Bi-based free-cutting brass had approximately the same fatigue performance as that of Pb-based free-cutting brass. It was also clarified that the addition of Bi or Pb initiated fatigue cracks, and that the crack growth period occupied most of the fatigue life. Differences in the FCG behavior of the three free-cutting brasses were observed in the low Δ*K* range. The modified linear fracture mechanics parameter *M* was used to quantitatively analyze the fatigue life and FCG behavior (short surface cracks). A comparison between the calculated and experimental results showed that *M* was useful.

## 1. Introduction

Brass is a widely used material for the manufacture of components, such as bolts, nuts, and valves, that are applied to products, such as watches, cameras, and musical instruments. Brass becomes hard by alloying soft copper (Cu) and zinc (Zn) and has acceptable mechanical properties. Depending on the mixing ratio of Cu and Zn, they are called 7-3 brass (α-brass) and 6-4 brass (α-β-brass). Furthermore, 6-4 Brass is suitable as a material for machine parts because of its low elongation and high tensile strength.

Free-cutting brass, which has high machinability, has been widely used [1]. The improved workability minimizes tool wear and allows the production of dimensionally precise parts with an excellent surface finish. Conventional free-cutting brass is an alloy of 6-4 brass in which 1.8–4.5 wt% lead (Pb) is added. However, because Pb is harmful, its use has been restricted in recent years [2]. Therefore, lead-free free-cutting brass has been developed in which bismuth (Bi) or silicon (Si) is added to brass instead of Pb [3]. Although Bi is a heavy metal, it has long been known to have few adverse effects on the human body and the environment.

When a Cu alloy is used as a mechanical/structural member, it is necessary to clarify the fatigue and fatigue crack growth (FCG) characteristics of the material. To date, several studies have been reported on the fatigue and FCG properties of Cu and Cu alloys.

First, there are several studies that have been conducted on 7-3 brass (α-brass). Beevers [4] studied the FCG behavior of 7-3 brass using three specimens with crystal grain sizes, *d*, of 51, 246, and 714 μm. Fatigue experiments were performed in a laboratory environment with a stress ratio *R* of 0.35 and a stress repetition rate *f* of 90 Hz. Consequently, the threshold value of the stress intensity factor range, Δ*K*_th_, increased with an increase in *d*. He reported that roughness-induced fatigue crack closure (RIFCC) was dominant in the FCG process of this material. Murakami et al. [5] studied the FCG properties of short cracks using 7-3 brass and clarified that the FCG properties of short cracks were equivalent to the Manson–Coffin law. Uematsu et al. [6] performed a plane bending fatigue test on 7-3 brass to study the effects of grain boundaries on the short-surface fatigue crack growth behavior in detail using atomic force microscopy (AFM). It has been reported that short surface-cracks occur along the slip system with the largest Schmid factor in the crystal grains. Additionally, crack branching and zigzag crack growth behavior were observed owing to cyclic work hardening during the fatigue process. Kaneshiro et al. [7] conducted a low-cycle fatigue experiment on 7-3 brass. They reported that fatigue cracks occurred at the grain boundaries because strain concentration points were generated in the crystal grains during fatigue.

Several studies have examined the FCG behavior of 6-4 brass (α-β brass). Murakami et al. [8] performed FCG experiments using several 6-4 brasses with different values of *d* to study the effects of *d* and *R* in the low-FCG velocity range. They reported that in the high FCG region, multiple slip systems operated simultaneously ahead of the fatigue crack tip before fatigue crack growth, indicating that the slip surface separation mechanism operated. They also reported striations on the fracture surfaces. In the low-FCG region, fatigue cracks were developed by a single slip system because the slip systems that could operate were limited. The fatigue fracture surface exhibited morphologies, such as cleavage, grain boundary facets, and striped patterns. Therefore, in the low-FCG region, the influence of *d* and *R* on the FCG velocity becomes significant. According to Sugeta et al. [9], in the low-FCG velocity range, owing to the effects of grain boundaries and repeated strain hardening, the slip system that had been operating until it was suppressed, and other slip systems began to operate, leading to zigzag FCG behavior. The above observations were similar to those for 7-3 brass [6]. Lin et al. [10] reported that the effect of *f* on the FCG properties of 6-4 brass materials was affected by *R*. It was also shown that the effect of specimen thickness on the FCG behavior was affected by the corrosive environment [11].

In recent years, studies have been conducted on the fatigue behavior of ultrafine grain Cu (hereinafter referred to as UFG Cu material) produced using an equal-channel angular pressing (ECAP) method [12]. Arzaghi et al. [13] studied the effect of *d* on the FCG properties of UFG Cu materials. They reported that in the low Δ*K* region, the FCG rate of the UFG Cu material was faster than that of normal Cu with coarse grains. This was because UFG Cu materials were less prone to RIFCC. Collini [14] studied the FCG behavior of a UFG Cu material with an average dimension of *d* = 300 nm. The FCG experiments were performed in an atmosphere with *R* in a range of 0.1–0.7. It has been reported that the FCG resistance of UFG Cu material was higher than that of coarse-grained material in the high Δ*K* region.

Goto et al. [15] studied the fatigue strength and FCG behavior (short surface cracks) in UFG Cu materials that were annealed and prepared by ECAP. Consequently, in the low and medium cycle fatigue regions, the annealing treatment reduced the fatigue life. However, in the high-cycle fatigue region, the fatigue strength of the UFG Cu material increased by approximately 9% after the annealing treatment. In addition, Goto et al. [16] conducted fatigue experiments on UFG Cu materials and observed surface damage during the fatigue process. They reported a close relationship between hardness changes and surface damage formation.

Although not directly related to the FCG behavior of brass, there are several studies on the occurrence and progression of damage during the fatigue process. Favier et al. [17] conducted ultra-high cycle fatigue experiments on α-brass and investigated the mechanism of microscopic damage generation in specimens. Skibicki et al. [18] conducted low-cycle fatigue experiments on CuZn37 brass and studied the stress-strain multiaxial fatigue criteria to predict fatigue life. Liu et al. [19] systematically investigated the relationship between the high-cycle fatigue behavior of 6-4 brass and the material microstructure. They studied the high-cycle fatigue failure mechanisms based on the evolution of dislocation structures. Furthermore, Liu et al. [20] investigated the high-cycle fatigue behavior of brass when subjected to laser shock peening (LSP) and the microstructural changes during the fatigue process. Still others, such as Sangeetha et al. [21], combined experimental and artificial neural network (ANN) methods to investigate the effect of specimen surface roughness on the fatigue life of brass and EN24 steel.

The following studies on fracture analysis and mechanical properties can be mentioned for free-cutting brass research. Chunlei et al. [22] performed a damage analysis of lead-free free-cutting brass valves installed in the piping of sanitary equipment. A leak was found adjacent to the brazing site, suggesting that the residual stress caused by excessive brazing and segregation of Bi caused valve breakage. Toulfatzis et al. [23] studied the mechanical behavior of free-cutting brass under static and dynamic loads. They showed that Pb-free brass could replace conventional Pb-containing brass.

As aforementioned, several studies have been conducted on the fatigue behavior of Cu and Cu alloys (pure Cu, 7-3 brass, 6-4 brass). However, many of these studies are concerned with the FCG behavior of long through-cracks, whereas there seems to be a lack of examples of the FCG behavior of short surface fatigue cracks.

Furthermore, no studies have been conducted on the fatigue behavior and FCG characteristics of free-cutting brass. This could be attributed to a greater emphasis on the development of lead-free free-cutting brass and the postponement of the accumulation of data on the strength and fatigue strength. When free-cutting brass is used in a variety of mechanical components, such as bolts, nuts, and valves, it is necessary to clarify the fatigue properties of the material and the effect of the microstructure on fatigue resistance. In this study, a tensile fatigue experiment was conducted at *f* = 15 Hz and *R* = 0.1 at room temperature using two types of Bi- and Pb-based free-cutting brasses with different *d* values. Subsequently, the fatigue strength, fatigue crack initiation, and FCG behavior of Pb- and Bi-based free-cutting brass materials were studied and clarified in detail.

## 2. Materials and Experimental Procedure

### 2.1. Materials

In this study, free-cutting brass containing Bi (JIS C6801, hereafter referred to as Bi material) and free-cutting brass containing Pb (JIS C3604, hereafter referred to as Pb material) were used as test materials. The test materials were prepared using a drawing process. Two types of Bi materials were prepared by adjusting the degree of drawing: a material with a large crystal grain size (Bi,_A_) and a material with a small crystal grain size (Bi,_B_).

#### 2.1.1. Microstructure of the Materials

After revealing the microscopic structure of the test material using an etching solution (7.5% NH_3_, 0.7% H_2_O_2_, 91.8% H_2_O, wt%), the microscopic structure of the material in the cross-section perpendicular to the drawing direction was observed using an optical microscope. Figure 1a–c showed the observation results for Bi,_A_, Bi,_B_, and Pb materials, respectively.

As shown in Figure 1a,b, the microscopic structure of Bi consisted of three phases: α, β, and Bi. In the Bi,_B_ materials, the phase boundaries of the α and β phases were bonded to mesh with each other and had a more complicated shape compared with that of the Bi,_A_ materials. Moreover, the α and β phases of the Bi,_B_ materials were smaller than those of Bi,_A_ materials. However, the microscopic structure of the Pb material in Figure 1c consisted of three phases: α, β, and Pb phases. The α phase was a face-centered cubic crystal of a Cu-Zn solid solution with good cold workability, and the β phase was an ordered nonstoichiometric intermetallic compound Cu-Zn, which had a body-centered cubic (bcc) lattice [23]. The cross-sectional grains exhibited an isotropic shape in all the materials and were not flat. Although the figure was omitted, the microscopic structure observed on the sample surface was stretched in the drawing direction.

For the component analysis of the α, β, Bi and Pb phases, an energy dispersive X-ray spectroscopy (EDX) that is installed in a scanning electron microscope (SEM) manufactured by JEOL Ltd. (JSM-7001F) was used because the phase size was only few tens of μm. The accelerating voltage was set at 15 keV. Figure 2 shows the SEM photograph of the Bi,_A_ materials, wherein the α, β, and Bi phases are indicated by arrows. Twenty measurement points were randomly selected for each phase, and point analysis was performed using EDX. Additionally, SEM was used to measure the microstructure dimensions by adopting a straight-line cutting method.

Table 1 lists the results of the EDX point analysis (average value at 20 points) and the average dimensions of each phase of the α, β, Bi, and Pb phases. From the table, all test materials contained approximately 60 wt% Cu and approximately 40 wt% Zn. The average crystal grain sizes of the α and β phases in the Bi,_A_ material were 6.9 and 4.5 μm, respectively, and the average crystal grain sizes of the α and β phases in the Bi,_B_ material were 5.4 and 3.9 μm, respectively. Therefore, the latter had a smaller crystal grain size than that of the former. In the Bi,_A_ and Bi,_B_ materials, most of the Bi was contained in the Bi phase, but it was also slightly contained in the α and β phases. Focusing on the Zn content of the Bi material, the Zn contents in the α and β phases were approximately 39 and 45 wt%, respectively. The Zn content of the β phase was higher than that of the α phase. 

In contrast, in the Pb material, the average crystal grain sizes of the α and β phases were 10.7 and 3.2 μm, respectively. The Zn contents in the α and β phases were approximately 38 and 45 wt%, respectively, and the Zn content of the latter was higher than that of the former. Most of the Pb was contained in the Pb phase, but it was also slightly contained in the α and β phases.

#### 2.1.2. Hardness of Material Microstructure

The micro Vickers hardnesses (HVs) of the α and β phases for the Bi,_A_ and Bi,_B_ materials were measured. A microhardness tester (Akashi Seisakusyo, Ltd., Akashi, Japan) was used for HV measurements. A test load of 10 gf was applied for 5 s for measurement. The HV measurements were performed at approximately 20 and 10 points for the α and β phases, respectively.

Figure 3a shows the Vickers indentation of the α phase in the Bi,_B_ material as an example. Indentations were placed approximately in the center of the α phase to avoid the effects of the grain boundaries. Figure 3b shows the HV measurement results for the α and β phases of Bi,_A_ and Bi,_B_ materials on a Weibull probability paper with HV on the horizontal axis and cumulative probability F(HV) on the vertical axis. The figure shows that the F(HV)−HV relationship was displayed with a two-parameter Weibull distribution because it was approximately a straight line on the Weibull probability paper.

The slope of the straight line for the α phase was greater than that for the β phase. In addition, the data variation in the α phase was smaller than that of the β phase. In addition, there was no difference between the Bi,_A_ and Bi,_B_ materials in the F(HV)−HV relationship. This was because (as listed in Table 1) there was no significant difference in the chemical compositions of the α and β phases of the two materials. The average hardness values of the α and β phases were HV 151 and HV 187, respectively, indicating that the latter was harder than that of the former. The hardness measurement results corresponded to the fact that the amount of Cu contained in the β phase was less than that of the α phase, and the amount of Zn contained in the β phase was more than that of the α phase (Table 1). This also corresponded to the fact that the α phase had a face-centered cubic lattice, and the β phase had a bcc lattice [23].

Since *d* was small and hardness measurement was difficult for the Bi phase, HV measurements were performed using a single-phase Bi sample. The hardness of Bi was HV 10. This value was considerably softer than that of the HV values in the α- and β-phases (HV 150–190). Moreover, the hardness of the Pb phase was not measured in this study. According to literature [24], it was approximately HV 4. Therefore, the addition of Bi and Pb to brass to introduce free-cutting properties was expected to reduce the strength of the material.

#### 2.1.3. Mechanical Properties of Test Materials

Table 2 lists the measurement results of the Young’s modulus *E*, yield strength σ_0.2_, tensile strength σ_B_, and elongation δ after conducting tensile tests on the three types of test materials. Round bar-shaped specimens, as shown in Figure 4a,b were used. The strain rate was 1.5 × 10^−3^ [s^−1^]. The tests were performed twice under the same conditions; the average values are listed in the table. From the table, the σ_0.2_ and σ_B_ values for the Bi,_B_ material were higher than those of the Bi,_A_, and Pb materials. Moreover, the differences between Bi,_A_ and Pb materials in the values of σ_0.2_ and σ_B_ were not significant.

### 2.2. Experimental Procedures

#### 2.2.1. Fatigue Specimen

Figure 4 shows the shapes and dimensions of the specimens used in this experiment to study the relationship between stress σ_max_ and fatigue life N_f_ (*S*-*N* curve). The specimen shown in Figure 4a was used for the Bi,_A_ and Pb materials, and the specimen shown in Figure 4b was used for the Bi,_B_ material. Since the Bi,_B_ material had a higher degree of drawing than that of the Bi,_A_ material, the specimen grip diameter of the former was narrower than that of the latter. However, the diameter (4 mm) and length (14–15 mm) of the parallel portions of both specimens were the same. In preliminary experiments, it was confirmed that there was no difference in the fatigue life owing to the difference in the shape of the specimen.

The specimen shown in Figure 4c was used for the FCG experiment. The observation area was narrowed to facilitate the measurement of the short surface fatigue crack. That is, a constriction with a depth of 0.25 mm and a radius of 12 mm was provided in the parallel portion of the smooth specimen. Additionally, it was designed so that short surface cracks would occur there. The stress concentration factor of the constricted part was 1.05.

The specimen surface was polished using SiC paper (No. 600–2000) and subsequently mirror-finished with diamond paste (1 μm) prior to the fatigue experiments.

#### 2.2.2. Fatigue Test

Using the specimens shown in Figure 4a,b, tensile fatigue tests with *R* = (σ_min_/σ_max_) = 0.1 were performed under a sinusoidal load waveform at *f* = 15 Hz. Here, σ_min_ and σ_max_ represent the minimum and maximum stress, respectively. The fatigue experiments were performed in a laboratory environment using an electric/hydraulic servo fatigue tester with a capacity of 20 kN. The fatigue fracture surfaces were observed using scanning electron microscopy (SEM; HITACHI, S-530).

#### 2.2.3. Fatigue Crack Length Measurement and Stress Intensity Factor Evaluation Formula

The fatigue experiments were performed under constant stress amplitudes (σ_a_) using the specimen shown in Figure 4c. The behavior of short-surface fatigue cracks generated and propagated on the specimen surface was investigated using the replica method [25]. The fatigue experiment was interrupted at regular intervals, and replicas of the specimen surfaces were collected using an acetyl cellulose film and methyl acetate (solvent). This series of operations were continued until the specimen failed. Fatigue crack initiation (FCI) and FCG behavior were investigated by tracing back from the replica just before failure to the replica at the beginning of the experiment using an optical microscope (200×).

The stress intensity factor, *K*, was used to evaluate the FCG properties of the surface cracks. To evaluate *K*, it was necessary to determine the shape of the surface crack. Therefore, *K* was calculated using Equation (1), assuming that the shape of the surface crack was semicircular.
(1)$$K=Y\sigma \sqrt{\pi a},$$
where *Y* was a correction coefficient, and *Y* = 0.73 [26] was used for the surface crack in semicircular shape, as described in Section 3.4.

## 3. Experimental Results

### 3.1. S-N Curves

Figure 5 shows the *S*-*N* curves (σ_max_–*N_f_* relationship) of the three types of free-cutting brass: Bi,_A_, Bi,_B_, and Pb materials at room temperature and in the atmosphere. From this figure, the experimental data (○) of the Bi,_A_ material were located approximately 20 MPa below those of the Bi,_B_ (△) and Pb (■) in the high-stress region of σ_max_ = 350 MPa or more, showing that the fatigue resistance of Bi,_A_ was lower than those of the Bi,_B_ and Pb materials. In addition, the *S*-*N* curves of the Bi,_B_, and Pb materials were approximately the same from the high to low-stress range; therefore, the fatigue resistance of the Bi,_B_ was comparable to that of the Pb material.

The three types of free-cutting brass exhibited smooth *S*-*N* curves with no break points from the high-stress regions to the low-stress regions. If the fatigue strength at *N_f_* = 2 × 10^6^ was defined as the fatigue limit σ_w_, the σ_w_ of the Bi,_A_ material is 320 MPa, whereas the σ_w_ of Bi,_B_ and Pb materials is 330 MPa; therefore, there are no significant differences between them.

### 3.2. Crack Initiation and Initial FCG Morphology

By continuous observation of the specimen surface, the initiation behaviors of fatigue cracks in free-cutting brass were investigated. As an example, Figure 6a,b compared the optical micrographs of Bi,_A_ before fatigue loading (*N* = 0) and at *N* = 10^4^ cycles, where *N* indicated the number of stress cycles. The fatigue experiment was performed under the conditions where σ_max_ = 330 MPa and *R* = 0.1. In these figures, the α, β, and Bi phases were indicated by the arrows. The largest Bi phase in the photograph was 717 μm^2^. At *N* = 10^4^ cycles (Figure 6b), as indicated by the red arrow, fatigue cracks occurred at the Bi/β and Bi/α interfaces and propagated inside the Bi phase. The black arrow indicated the crack tip. Although the figure was omitted, similar results were observed for the Bi,_B_ and Bi,_A_ materials. In the Pb material, the Pb/β and Pb/α interfaces were the crack initiation sites.

Figure 6c shows an SEM image of the fracture surface of the Bi,_A_ material in the same experiment as in Figure 6a,b. The photograph was taken by tilting the specimen so that both the surface and the fractured surface of the specimen could be seen. The point indicated by the white arrow in the figure represents the fatigue crack-initiation site. As shown in Figure 6c, fatigue cracks are initiated at the interface of the Bi phase (light gray). In addition, the appearance of the fatigue fracture surface, including the crack initiation site, was severely rough. Further, as shown by the black arrows, the fracture surface had morphologies, such as grain boundary facets and striped patterns related to the microscopic structure of the material. These results were similar to those of Beevers [4] and Murakami et al. [8].

### 3.3. Short Surface FCG Behavior 

Figure 7 shows the FCG morphology of surface cracks during the fatigue process of the Bi,_A_ material. The experimental conditions were as follows: room temperature, σ_max_ = 350 MPa, *R* = 0.1, and *f* = 15 Hz. Figure 7a shows an optical micrograph of the fatigue crack transferred to a replica. It was shown at low magnification to understand the macroscopic FCG morphology. At *N* = 2.5 × 10^5^, multiple cracks A, B, and C initiated and grew, as indicated by the arrows in the figure. At *N* = 4.2 × 10^5^, multiple cracks A, B, and C grew in a zigzag manner under the influence of the microscopic structure of the material. In addition, cracks with a length of 1 mm or greater grew by coalescing with other cracks generated at other locations.

Subsequently, the microscopic FCG morphology of crack C, shown in Figure 7a, was continuously observed from *N* = 3 × 10^4^ to 2.5 × 10^5^. The results are shown in Figure 7b. At *N* = 3 × 10^4^, two short surface cracks ① and ② with a length of approximately 20 μm occurred. The arrows in the figure indicate the tips of the short surface cracks. From *N* = 5 × 10^4^ to 1 × 10^5^, these short surface cracks grew diagonally rather than perpendicularly to the loading direction. At *N* = 2.5 × 10^5^, cracks ① and ② coalesced into a long crack of approximately 120 μm. The short surface cracks grew in a zigzag manner because the cracks grew along the α/Bi and α/β grain boundaries. The FCG morphology of such short surface cracks was similar to the observations of 7-3 [6] and 6-4 brasses [9].

### 3.4. FCG Behavior of Short Surface Cracks 

Figure 8 shows the log (2*a*) − *N* relationship at room temperature for Bi,_A_, Bi,_B_, and Pb materials, where 2*a* is the total crack length. The crack length projected onto a plane perpendicular to the loading direction was measured as 2*a*. For the Bi,_A_ material, the FCG behavior was observed at σ_max_ = 350 and 400 MPa using two specimens. At σ_max_ = 350 MPa, 2*a* was measured for cracks A, B, and C, and at 400 MPa, 2*a* was measured for cracks A and B. From the figure, the log (2*a*) − *N* relationship was approximated by an approximately straight line in the region where 2*a* was 50 μm or greater, excluding the unstable FCG region. Similar results were reported by Goto et al. [16] for UFG Cu. These were characteristics of microcrack growth [27].

Focusing on the difference in free-cutting brasses between the Bi,_A_, Bi,_B_, and Pb materials, at σ_max_ = 350–370 MPa, the values of *N* required to grow to 2*a* = 5 × 10^−4^ m were *N* = 4 × 10^5^ for Bi,_A_, and approximately *N* = 8 × 10^5^ for the Pb and Bi,_B_ materials. Therefore, of the three types of free-cutting brass, the Bi,_A_ had the fastest FCG speed, followed by Pb and Bi,_B_ materials. The tendency of this FCG behavior was consistent with that in the *S*−*N* curve. In addition, in all materials, fatigue cracks occurred at the initial stage of the fatigue life (*N*/*N_f_* = 0–10%), and most of the fatigue life was occupied by the FCG process.

Figure 9 shows an example of the fracture surface of the Bi,_A_ specimen (SEM photograph) which broke in air at room temperature. The experimental conditions were σ_max_ = 330 MPa and *N*_f_ = 7 × 10^5^. The fatigue surface crack originated from the crack origin (yellow arrow) and grew downward, as depicted in the figure. The multiple white arrows in the figure indicate the shapes of the surface cracks. The length/depth ratio of the surface crack was approximately 0.49. At this point, crack length 2*a* on the specimen surface was approximately 200 μm. Although the figure has been omitted, the same observation results as aforementioned were obtained for other surface cracks as well. Therefore, *K* was calculated using Equation (1) for the surface crack shape, that is, in a semicircular shape.

Approximating the 2*a* − *N* relations in Figure 8 with curves and using these curves, Δ2*a* (increment of 2*a*) was obtained with respect to Δ*N* (increment of *N*). The FCG velocity (d*a*/d*N* ≅ Δ*a*/Δ*N*) was subsequently calculated. By substituting *a* (at 0.5 × Δ*N*) and σ_max_ into Equation (1), *K*_max_ was obtained and Δ*K* was obtained with *R* = 0.1.

Figure 10 shows the d*a*/d*N* − Δ*K* relationships of the short surface cracks of three types of free-cutting brasses (Bi,_A_, Bi,_B_, and Pb) obtained at *R* = 0.1. For comparison, the d*a*/d*N* − Δ*K* relationships [8] obtained for the long penetrating cracks of 6-4 brass (*d* = 15 μm, σ_B_ = 430 MPa, HV 77) were also shown in this figure. These relationships were obtained for three different *R* values (0.06, 0.3, and 0.6) using the CT specimens.

Focusing on the difference between the three types of free-cutting brasses with regards to FCG behavior, differences in d*a*/d*N* between materials appeared in the low Δ*K* region below Δ*K* ≅ 4–5 MPam^1/2^. That is, d*a*/d*N* at a constant Δ*K* was faster in the order of Bi,_A_, Pb, and Bi,_B_ materials. This result corresponded to the measurement results of the 2*a* − *N* relationship shown in Figure 8. However, in the high d*a*/d*N* region where Δ*K* was larger than 4–5 MPam^1/2^, there was no significant difference in the d*a*/d*N* of the three types of free-cutting brasses.

In the fracture surface shown in Figure 9, in the region where the crack depth was less than 100 μm (Δ*K* ≅ 4–5 MPam^1/2^), the fracture surface showed severe unevenness and rough properties. In contrast, the fracture surface in the region deeper than 100 μm showed relatively flat properties. Therefore, the region where the FCG behavior differed between the materials was the region where the fracture surface was rough, and the FCG behavior was sensitive to the microscopic structure of the material.

Furthermore, in the low Δ*K* region of the Bi,_A_ and Pb materials, d*a*/d*N* at σ_max_ = 400 MPa was larger than that at 350 MPa at a constant Δ*K* value. This stress dependence on the d*a*/d*N* − Δ*K* relationship indicated that it was difficult to apply linear elastic fracture mechanics to the FCG behavior of short surface cracks [5,27].

Focusing on the difference between short surface cracks and long penetrating cracks in the FCG behavior, in the high Δ*K* range (Δ*K* was greater than 4−5 MPam^1/2^), the FCG velocities of both were approximately the same. However, in the low Δ*K* region (Δ*K* was smaller than 4–5 MPam^1/2^), the Δ*K*_th_ for short surface cracks was lower than that for penetrating long cracks. 

It was reported that, in the high Δ*K* region (d*a*/d*N* faster than 5 × 10^−9^ (m/cycle)) of long through-cracks, multiple slips occurred owing to fatigue loading, and fatigue cracks occurred along the slips [8]. Subsequently, the fatigue crack propagated owing to the slip-surface separation mechanism, and there were also several striations on the fracture surface [8]. 

The differences in the FCG behavior between long and short surface cracks were discussed quantitatively in Section 4.2.

### 3.5. FCG Path 

From Figure 10, the FCG velocity of the Bi,_A_ material was higher than that of the Bi,_B_ material in the low Δ*K* region. In this section, we considered the reasons for this finding. The surfaces of Bi,_A_ and Bi,_B_ materials were etched to reveal the α, β, and Bi phases. FCG path experiments were performed using four specimens at σ_max_ = 340 and 350 MPa for the Bi,_A_ material and at σ_max_ = 360 and 370 MPa for the Bi,_B_ material, respectively. 

As shown in Figure 3b, the β phase was harder than that of the α phase; therefore, we focused on the ratio of the β phase to the FCG path of each material.

An explanation of the FCG path is shown in Figure 11a. In this figure, red virtual fatigue cracks, symbols ① and ②, and the α and β phases (arrows) are shown. ① indicates the location where the crack had propagated in the β phase grain, and ② corresponded to the location where the crack had propagated between the β and α phases (β/α interface).

To quantitatively study the crack growth path, *ϕ*_β+β/α_, *ϕ*_β_, and *ϕ*_β/α_, defined by Equation (2), were introduced. In Equation (2), ∑ represents the summation symbol; *i* represents the number of relevant parts; *ϕ*_β + β/α_ is the ratio at which the crack propagates within the β phase and along the β/α interface; *ϕ*_β_ is the ratio at which the crack propagates within the β phase; and *ϕ*_β/α_ is the ratio at which the crack propagates along the β/α interface.
(2)$$\phi_{\beta +\beta /\alpha }=\sum \left({①}_{i}+{②}_{i}\right)/2a,\;\phi_{\beta }=\sum {①}_{i}/2a,\;\phi_{\beta /\alpha }=\sum {②}_{i}/2a,$$

Figure 11b–d show variations in *ϕ*_β+β/α_, *ϕ*_β_, and *ϕ*_β/α_, respectively, as a function of Δ*K*. The experimental data points in these figures were the averages of the observational results obtained from the two experiments. From Figure 11b, in the region where Δ*K* was less than 4 MPam^1/2^, *ϕ*_β+β/α_ of the Bi,_B_ material was approximately 27–40%, which was 1.5–2 times larger than that (20%) of the Bi,_A_ material. Furthermore, from Figure 11c, in the region where Δ*K* was less than 4 MPam^1/2^, *ϕ*_β_ of the Bi,_B_ material was approximately 20%, which was clearly larger than that of the Bi,_A_ material (approximately 5–10%). In Figure 11d, the difference between the Bi,_A_ and Bi,_B_ materials with regards to *ϕ*_β/α_ was small overall.

From the aforementioned experimental results, in the Bi,_B_ materials, cracks had a higher ratio of the β-phase-related crack path as compared with the Bi,_A_ materials.

## 4. Discussion

### 4.1. Fatigue Crack Initiation Mechanism and FCG Behavior in the Low ΔK Range

As described in Section 3.2, in the Bi,_A_ and Bi,_B_ materials, fatigue cracks occurred at the Bi/β and Bi/α phase boundaries. In Pb, fatigue cracks occurred at the Pb/β and Pb/α phase boundaries. In this section, the crack generation mechanism in free-cutting brasses is considered.

As shown in Figure 3b, the HV values of the α and β phases were 150–190, whereas those of the Bi and Pb phases were approximately HV 10. Therefore, at the Bi/β and Pb/β phase boundaries, strong inhomogeneity occurs owing to the large difference in the hardness values of each phase. As a result, it can be inferred that stress and strain concentrations occur in the vicinity of the phase boundary, where cracks occur.

To calculate the stress concentration at the Bi/β phase boundary, an analysis using the finite element method (FEM) was performed using a model in which a Bi hemisphere was embedded in the surface of the β phase semi-infinite body; the Bi phase had an irregular shape, but for simplicity it was approximated to a sphere. The commercially available software ANSYS was used for FEM analysis. The values of Young’s modulus *E* for the β and Bi phases were set at 96 and 31.9 GPa, respectively. For the *E* value of Bi phase, the value from the literature was used [28]. Since this was a simple calculation, the details of the FEM analysis were omitted. The stress concentration factor at the Bi/β interface took a value of 1.49 and converged to one as the distance from the interface increased. Therefore, it is presumed that stress concentrations occur at the Bi/β or Bi/α interfaces owing to the significant difference in HV at the phase boundary, and cracks are generated at these interfaces.

Hatanaka et al. [29] reported that in Cu and 7-3 brass (α-brass), slips generated within the crystal grains accumulate at the grain boundaries, causing fatigue cracks. Similar results have been reported for Cu and Cu alloys [5,7]. Therefore, in free-cutting brass, the fatigue crack generation mechanism is different from that of other Cu alloys because Bi or Pb that are added to improve the free-cutting property cause cracks. 

Regarding the FCG behavior in the low Δ*K* region, the difference between Bi,_A_ and Bi,_B_ materials was considered. As shown in Figure 1, the α/β-phase boundaries in the Bi,_B_ materials are more strongly bonded (in comparison to the Bi,_A_ materials) and intermesh with each other. In the low Δ*K* region of the Bi,_B_ material, fatigue cracks tend to avoid the strong α/β-phase boundaries (Figure 11d) and propagate inside the β phase (Figure 11c). Consequently, the FCG velocity of the Bi,_B_ material is lower than that of the Bi,_A_ material.

The aforesaid differences between the Bi,_A_ material and the Bi,_B_ material are caused by the degree of drawing. Determination of methods to strengthen the bonding of the α/β-phase boundaries by either adjusting the degree of work or other alternative methods requires further research.

### 4.2. Derivation of the S−N Curve Based on the Modified Linear Elastic Fracture Mechanics (MLEFM) Parameter M

From the stress amplitude dependence of the d*a*/d*N* − Δ*K* relationship in Figure 10, linear elastic fracture mechanics (LEFM) cannot be applied to the FCG behavior of short surface fatigue cracks. Therefore, the FCG behaviors of short surface fatigue cracks were analyzed using the modified linear elastic fracture mechanics (MLEFM) parameter *M* [26,27,30].

The FCG relationship represented in Equation (3) was used as the basic equation for the analysis [31,32,33].
(3)$$\frac{da}{dN}=A{\left(\Delta K_{\mathrm{eff}}-\Delta K_{\mathrm{effth}}\right)}^{2}$$
where *A* is a material constant; and Δ*K*_eff_ (= *K*_max_ − *K*_op_) and Δ*K*_effth_ are the effective stress intensity factor range and threshold of the effective stress intensity factor range, respectively; *K*_max_ and *K*_op_ are the maximum stress intensity factor and the crack opening stress intensity factor, respectively. For analyzing the FCG behavior of short-surface cracks, Equation (3) was modified for the following three reasons [26,27,30]:(a)Elastoplastic effect: in the FCG process of short cracks, the FCG behavior does not satisfy the LEFM condition because the size of the plastic region, *R*_p_, formed at the crack tip is larger than that of the crack length. Irwin [34] proposed using a modified crack length (*a*_mod_) to incorporate the elastoplastic effect into the LEFM. *a*_mod_ was defined as the actual length plus half of *R*_p_. Using Dugdale’s equation [35] to calculate *R*_p_, *a*_mod_ is given by the following:


(4)$$a_{\mathrm{mod}}=a+\frac{1}{2}\left\{\sec\left(\frac{\pi }{2}\frac{\sigma_{\max}}{\sigma_{Y}}\right)-1\right\}a=aF,\;F=\frac{1}{2}\left\{\sec\left(\frac{\pi }{2}\frac{\sigma_{\max}}{\sigma_{Y}}\right)+1\right\},$$
where *F* is the elastic-plastic correction factor; and σ_Y_ represents the yield strength of the material.

In Dugdale’s *R*_p_ analysis, he assumed a long, slender plastic zone at the crack tip in a nonhardening material under plane stress. However, the free-cutting brass used in this study exhibits work-hardening. As such, Dugdale’s expression for *R*_p_ could have overestimated the size of the plastic zone in free-cutting brass. In addition, σ_Y_ of the free-cutting brass before the fatigue load shows a low value; however, it is considered to increase owing to work-hardening due to repeated stress. Therefore, in this study, the value of σ_B_ was used as that of σ_Y_.

(b)Fatigue crack closure (FCC): in short fatigue cracks, *K*_op_ increased from zero to *K*_opmax_ as the amount of crack growth increased. Here, *K*_opmax_ indicated the saturation value of the FCC, which was the *K*_op_ value for long-penetrating cracks. The change in *K*_op_ with crack growth was approximated using the following exponential function [26,27]:

(5)$$\Delta K_{\mathrm{op}}=\left(1-e^{-k\lambda }\right)\left(K_{\mathrm{opmax}}-K_{\min}\right),$$
where *k* is a parameter related to the rate of increase in *K*_op_, and its value differs depending on the material; λ is the amount of crack growth, and is calculated by using λ = *a*−*r*_e_, where *r*_e_ is the initial crack length (see condition (c)).

(c)According to Kitagawa et al. [36], the lower limit condition of the FCG for short fatigue cracks was defined by the fatigue limit Δσ_w_ rather than the threshold Δ*K*_effth_ for long penetrating cracks. To incorporate this condition, the material constant, *r*_e_ defined in Equation (6) was introduced [26,27] using the following:
(6)$$r_{e}={\left(\frac{\Delta K_{\mathrm{effth}}}{\Delta \sigma_{w}}\right)}^{2}\left\{\frac{1}{2\pi F\left(1+\sqrt{2}Y+0.5Y^{2}\right)}\right\}.$$

When the aforementioned three modifications (a), (b), and (c) were applied to Equation (3), the following equation was obtained, which provides the FCG velocity of a short fatigue crack:(7)$$\frac{da}{dN}=A{\left[\left(\sqrt{2\pi r_{e}F}+Y\sqrt{\pi aF}\right)\Delta \sigma -\left(1-e^{-k\lambda }\right)\left(K_{\mathrm{opmax}}-K_{\min}\right)-\Delta K_{\mathrm{effth}}\right]}^{2}.$$

The first term on the right side of Equation (7) corresponded to condition (c), the second term corresponded to condition (a), and the third term corresponded to condition (b). When the inside of the parentheses on the right side of Equation (7) was expressed using the MLEFM parameter *M*, the following equation was obtained: (8)$$\frac{da}{dN}=AM^{2}.$$

To perform the analysis using Equations (7) and (8), it was necessary to obtain material constants, such as Δ*K*_effth_, *K*_opmax_, and *k*, for long-penetration fatigue cracks in the equation. To the best of our knowledge, the aforementioned material constants for Cu and Cu alloys have not been reported.

A study on the FCG behavior of 7-3 brass by Beevers [4] reported that Δ*K*_th_ increased with increasing *d* owing to RIFCC. Similar results have been reported by Murakami et al. [8].

Ishihara et al. [37] studied the FCC properties of carbon steel and aluminum alloys during the FCG process and clarified that carbon steels were dominated by RIFCC and aluminum alloys were dominated by plastic-induced crack closures (PIFCC). Due to the *k* value of carbon steel, which indicated RIFCC was 6000 (m^−1^) [37], this *k* value was applied to the free-cutting brass used in this study.

The PIFCC behavior is affected not only by the thickness of the specimen, but also by the level of Δ*K* [38] and the work-hardening ability of the materials [39,40]. Since the brass specimen in this test has a relatively high work-hardening property, it can be inferred that it is difficult for the PIFCC to occur and, instead, the RIFCC behavior should be expected. A detailed examination of this aspect will be a subject for future research.

Several studies have been conducted on the FCG properties of carbon steels, and it has been reported that the value of Δ*K*_effth_ was approximately 3 MPam^1/2^ [26,41]. It has also been reported that the FCG properties of several materials, including steels, aluminum alloys, and Ti alloys, were unified by the d*a*/d*N* − Δ*K*_eff_/*E* relationship [42,43]. Assuming that the Δ*K*_effth_/*E* values for carbon steel and free-cutting brass were equal in consideration of the aforementioned previous research results, the Δ*K*_effth_ value for free-cutting brass was estimated to be approximately 1.5 MPam^1/2^. The value of *K*_opmax_ was set by referring to the d*a*/d*N* − Δ*K* relationship (*R* = 0.6) [8] obtained using 6-4 brass. In other words, by assuming that the *K*_min_ value [8] (4.8–5.2 MPam^1/2^) in the low Δ*K* range was larger than that of the *K*_opmax_ value, therefore, *K*_opmax_ = 4 MPam^1/2^. 

Table 3 lists the material constants. For simplicity, it was assumed that these values were the same for specimens Bi,_A_, Bi,_B_, and Pb. However, from the *S*-*N* diagram in Figure 5, the Δσ_w_ values for each specimen were different. Therefore, *r*_e_ had a different value (see Equation (6)), depending on the specimen.

Figure 12 shows the d*a*/d*N* − *M* relations for the Bi,_A_ (Figure 12a), the Bi,_B_, and the Pb materials (Figure 12b) on a log-log graph. In Figure 12b, the data for the Bi,_A_ material were plotted for comparison. Although there was a data scatter, the d*a*/d*N* − *M* relationship for the Bi,_A_ material (Figure 12a) was approximated by a straight line with a slope of two on the log-log graph. Similar results were also observed in Figure 12b, which showed the data for the Bi,_B_, and Pb materials. The straight lines in the figures were expressed using Equations (9) and (10), respectively.
(9)$$\frac{da}{dN}=3.5\times 10^{-11}M^{2},(\mathrm{for}\;\mathrm{Bi}{,}_{A}),$$
(10)$$\frac{da}{dN}=2\times 10^{-11}M^{2},(\mathrm{for}\;\mathrm{Bi}{,}_{B},\;\mathrm{and}\;\mathrm{Pb}).$$

In the d*a*/d*N* − *M* relationship, the difference between the two σ_max_ values of 350 and 400 MPa was small and approximately disappeared. This result is different from the d*a*/d*N* − Δ*K* relationship (Figure 10), which shows a σ_max_ dependence.

Therefore, the *M* parameter is effective for the analysis of FCG behavior of short surface fatigue cracks. 

A detailed observation of Figure 12b revealed that the d*a*/d*N* values of Bi,_B_, and Pb were approximately the same. In addition, in the region below *M* = 4 MPam^1/2^, the d*a*/d*N* of the Bi,_A_ material was faster than that of the Bi,_B_, and Pb materials, indicating that the FCG resistance of the Bi,_B_, and Pb materials was higher than that of the Bi,_A_ materials. This is because, in the Bi,_B_ materials, fatigue cracks propagate more frequently within the β phase than in the Bi,_A_ materials (Figure 11b,c). The hardness of the β phase (HV 187, body-centered cubic lattice) was higher than that of the α phase (HV 151, face-centered cubic lattice), and their crystal structures were different. Therefore, it is assumed that the FCG resistance of the β phase is higher than that of the α phase.

The slope of the d*a*/d*N* − *M* relationship for Bi,_B_, and Pb materials was larger than the slope of two of the straight lines shown in the figure. The reason for this seems to be that with the calculation of *M*, the same parameter values were used for the three free-cutting brasses.

The FCG lifetime *N*_p_ was obtained by integrating d*N* from *N*_i_ to *N*_f_, as shown in Equation (11). Here, *N*_i_ was the crack-initiation life. Furthermore, the integral for d*N* was converted into the integral for d*a* from the initial half-length from *r*_e_ to *a*_f_ using the FCG rule of Equations (9) and (10), respectively. Here, *a*_f_ was the half-length of the crack when the specimen broke, and *a*_f_ = 2 mm (total length of 4 mm) was set in this study, where *N*_f_ was the sum of *N*_i_ and *N*_p_ (*N*_f_ = *N*_i_ + *N*_p_). As shown in Figure 7 and Figure 8, fatigue cracks occurred early in the fatigue process (*N*_i_/*N*_f_ = 5%); therefore, *N*_i_ was negligible and was approximated as *N*_f_ ≅ *N*_p_ as shown in Equation (11).
(11)$$N_{f}\;≅N_{p}\;=\int_{N_{i}}^{N_{f}}dN\;=\int_{r_{e}}^{a_{f}}\frac{da}{AM^{2}}\;.$$

The *S*−*N* curves for Bi,_A_, Bi,_B_, and Pb materials were calculated by numerically integrating Equation (11). Simpson’s rule was used for the numerical integration. In Figure 13, the calculation and experimental results were compared. The curves (solid lines) in the figures show the calculated results. The calculated results and the experimental data are correlated.

Thus, the method of analyzing the FCG behavior of short surface fatigue cracks using *M* and calculating the *S*-*N* curve based on the analysis results is a useful analysis tool.

## 5. Conclusions

Axial tensile fatigue experiments were conducted at room temperature under a condition of *R* = 0.1 using three types of free-cutting brasses (Bi,_A_, Bi,_B_, and Pb materials) to study the fatigue life, FCI, and FCG characteristics. The results are summarized as follows: (1)Bi-based free-cutting brass had approximately the same fatigue performance as Pb-based free-cutting brass; therefore, it was expected to be used as a lead-free free-cutting brass in the future. More specifically, the fatigue resistance of lead-free Bi,_A_ was lower than that of Bi,_B_ and Pb materials.(2)Fatigue cracks occurred in less than 10% of the fatigue life, and most of the fatigue life was occupied by the crack-growth process. In the Bi,_A_ and Bi,_B_ materials, fatigue cracks were initiated along the Bi/β boundary or the Bi/α boundary. However, in the Pb material, fatigue cracks occurred along the Pb/β or Pb/α phase boundaries. This was because the hardness values of the Bi and Pb phases were considerably lower than those of the α and β phases, so that stress concentration occurred at the phase boundary.(3)The difference between the three types of free-cutting brass in the FCG behavior appeared in the low Δ*K* region where Δ*K* ≅ 4−5 MPam^1/2^ or less. The FCG velocity at a constant Δ*K* increased in the order of Bi,_A_, Pb, and Bi,_B_ materials. However, in the high d*a*/d*N* region where Δ*K* was 4–5 MPam^1/2^ or more, there was no difference in the d*a*/d*N* of the three types of free-cutting brasses. In addition, the d*a*/d*N* − Δ*K* relationship had a stress amplitude dependence, which made it difficult to apply the LEFM.(4)In the low Δ*K* region, the FCG resistance of the Bi,_B_ was higher than that of the Bi,_A_ material. The reason for this was that in the Bi,_B_ material, the frequency of fatigue cracks developing in β-phase grains (HV 187) was higher than that in the Bi,_A_ material.(5)The FCG behaviors of short surface fatigue cracks in free-cutting brass were analyzed using the MLEFM parameter *M*. Since the *S*-*N* curve calculated based on the analysis results was consistent with the experimental results, *M* was effective for the analysis of the fatigue life and FCG behavior.

## Figures and Tables

**Figure 1 materials-15-07488-f001:**
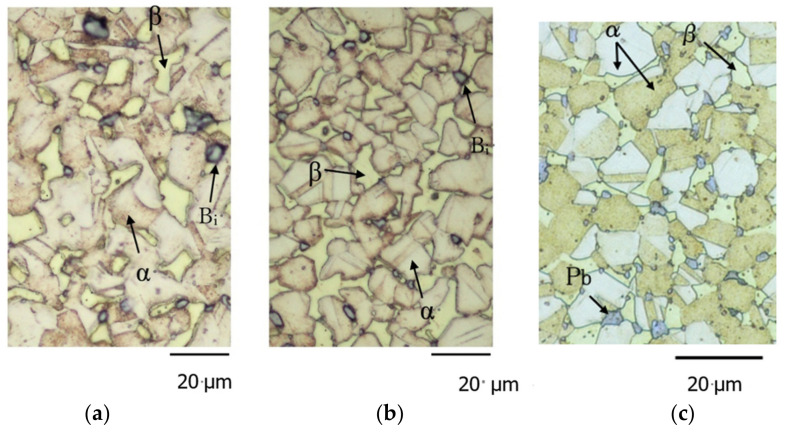
Microscopic structure of the materials used in this study: (**a**) Bi,_A_; (**b**) Bi,_B_; (**c**) Pb.

**Figure 2 materials-15-07488-f002:**
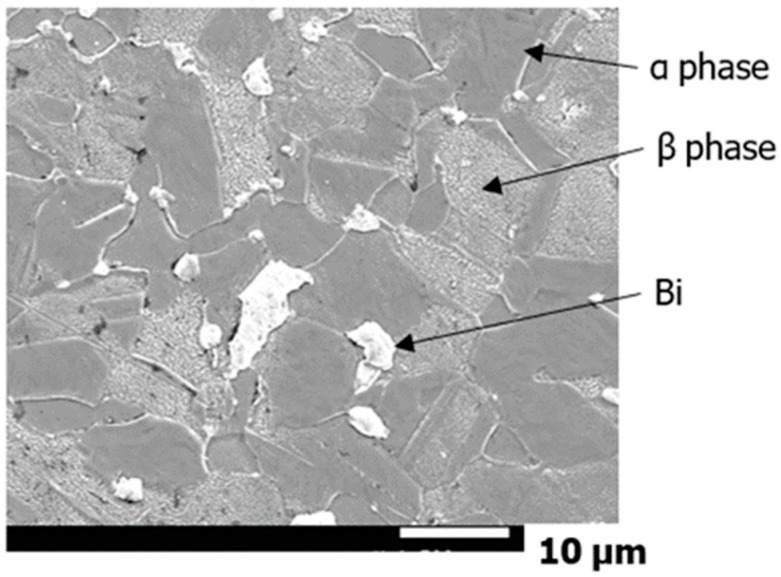
Bi,_A_ material microstructure (SEM).

**Figure 3 materials-15-07488-f003:**
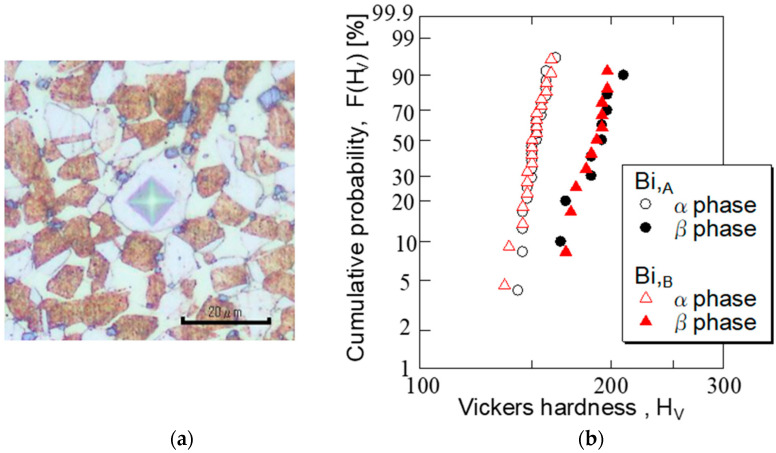
Statistical distribution of HV in the α and β phases. (**a**) Indentation morphology (α-phase of Bi,_B_); (**b**) statistical distribution of HV (Bi,_A_ and Bi,_B_ materials).

**Figure 4 materials-15-07488-f004:**
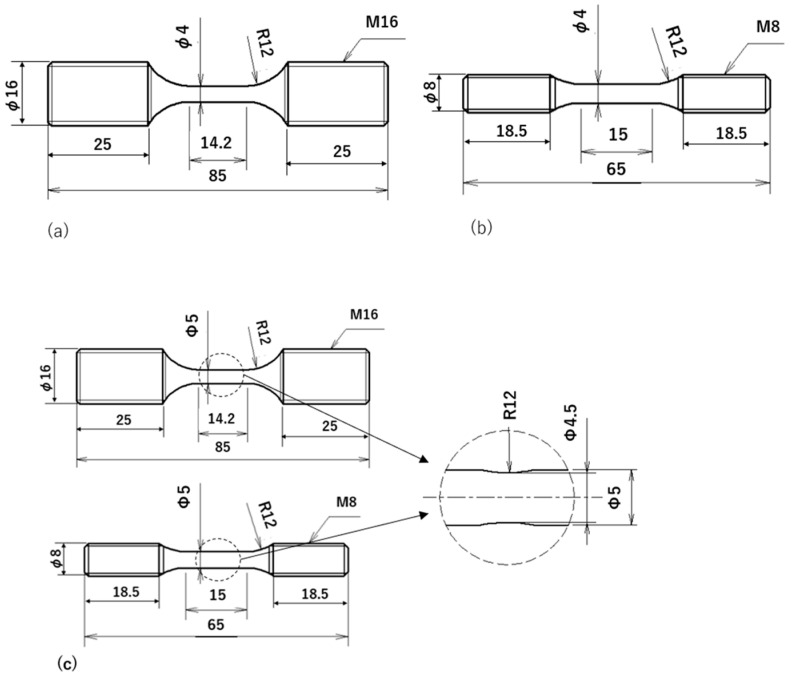
Shape and dimensions of specimens (mm): (**a**) Bi,_A_ and Pb materials; (**b**) Bi,_B_ material; (**c**) and FCG tests.

**Figure 5 materials-15-07488-f005:**
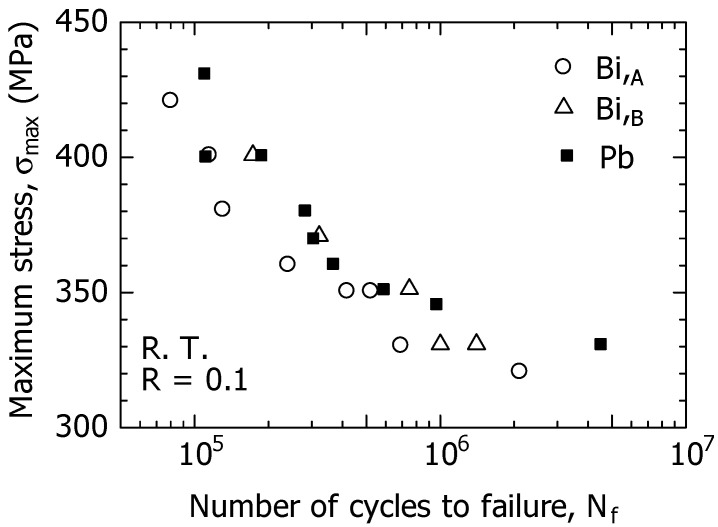
σ_max_–*N_f_* relations for Bi,_A_, Bi,_B_, and Pb materials at *f* = 15 Hz and *R* = 0.1.

**Figure 6 materials-15-07488-f006:**
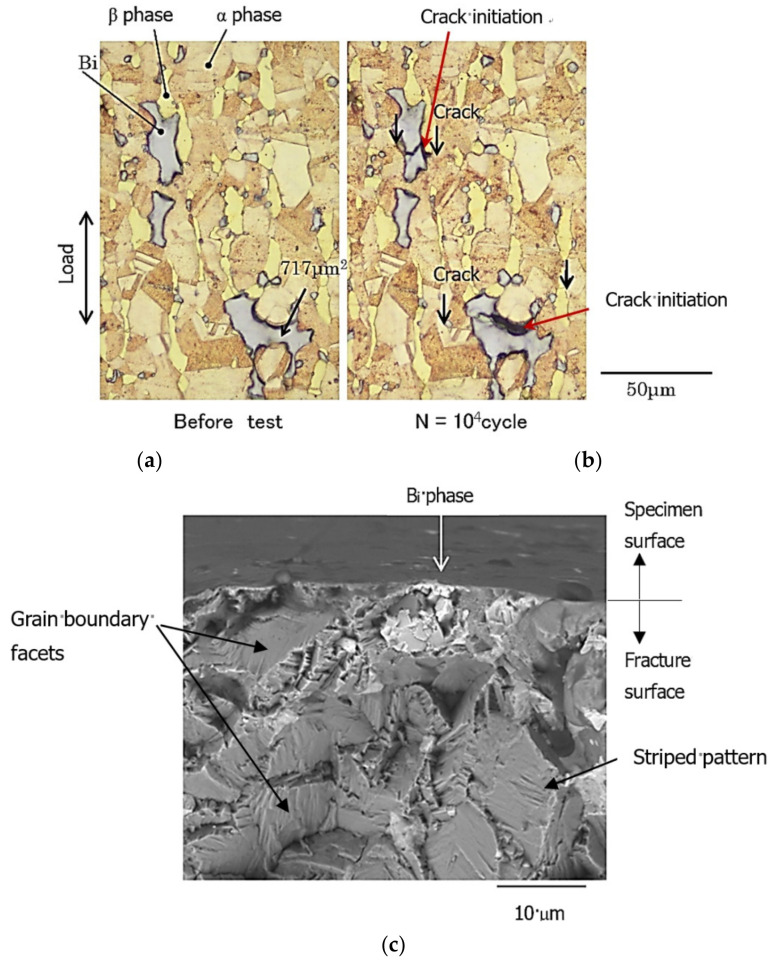
Fatigue crack initiation behavior observed in the Bi,_A_ material’s surfaces at σ_max_ = 330 MPa and *R* = 0.1: (**a**) specimen (*N* = 0); (**b**) specimen (*N* = 10^4^ cycles); and (**c**) fracture surface.

**Figure 7 materials-15-07488-f007:**
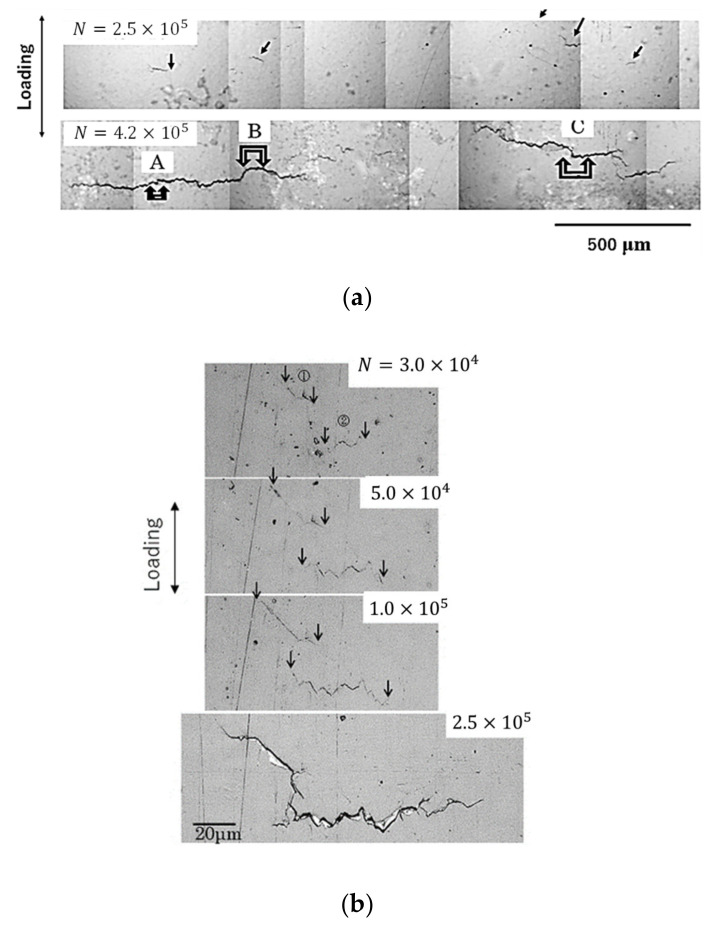
FCG behavior (Bi,_A_ material, σ_max_ = 350 MPa, *R* = 0.1) at different magnifications: (**a**) low and (**b**) high (crack C).

**Figure 8 materials-15-07488-f008:**
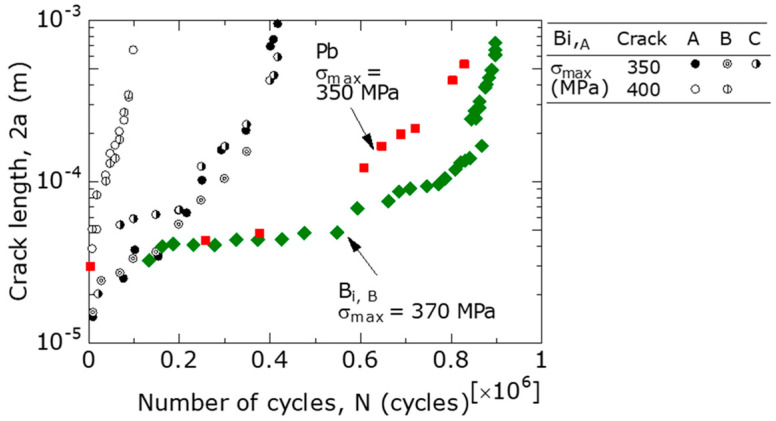
FCG curves at *R* = 0.1.

**Figure 9 materials-15-07488-f009:**
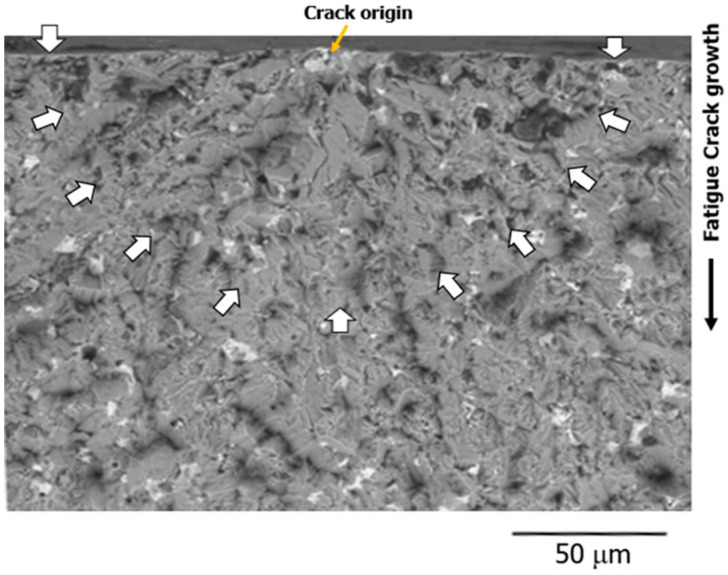
A semicircular surface crack observed on the fracture surface. (Bi,_A_ material, σ_max_ = 330 MPa, *N*_f_ = 7 × 10^5^).

**Figure 10 materials-15-07488-f010:**
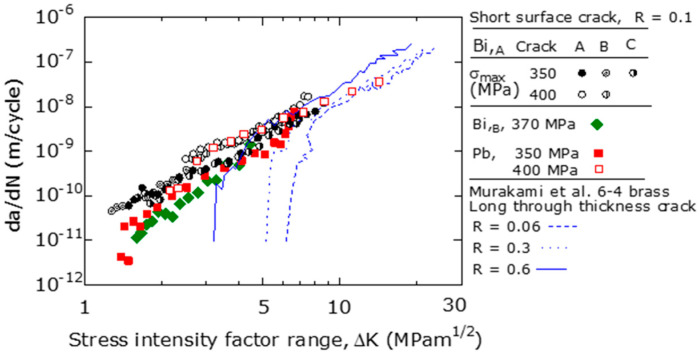
d*a*/d*N* − Δ*K* relationships for three types of brasses (*R* = 0.1).

**Figure 11 materials-15-07488-f011:**
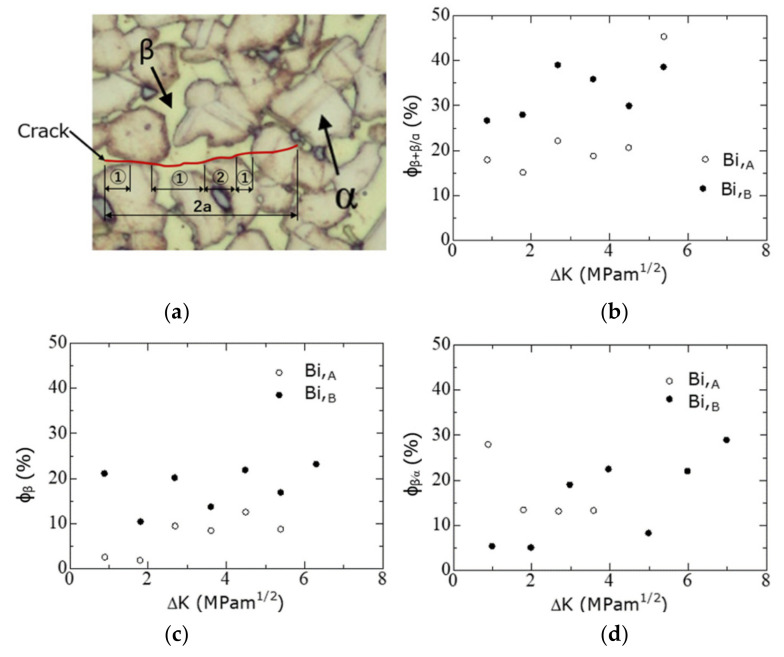
Involvement of β phase in the FCG path: (**a**) illustration of FCG path; (**b**) *ϕ*_β+β/α_; (**c**) *ϕ*_β_; and (**d**) *ϕ*_β/α_.

**Figure 12 materials-15-07488-f012:**
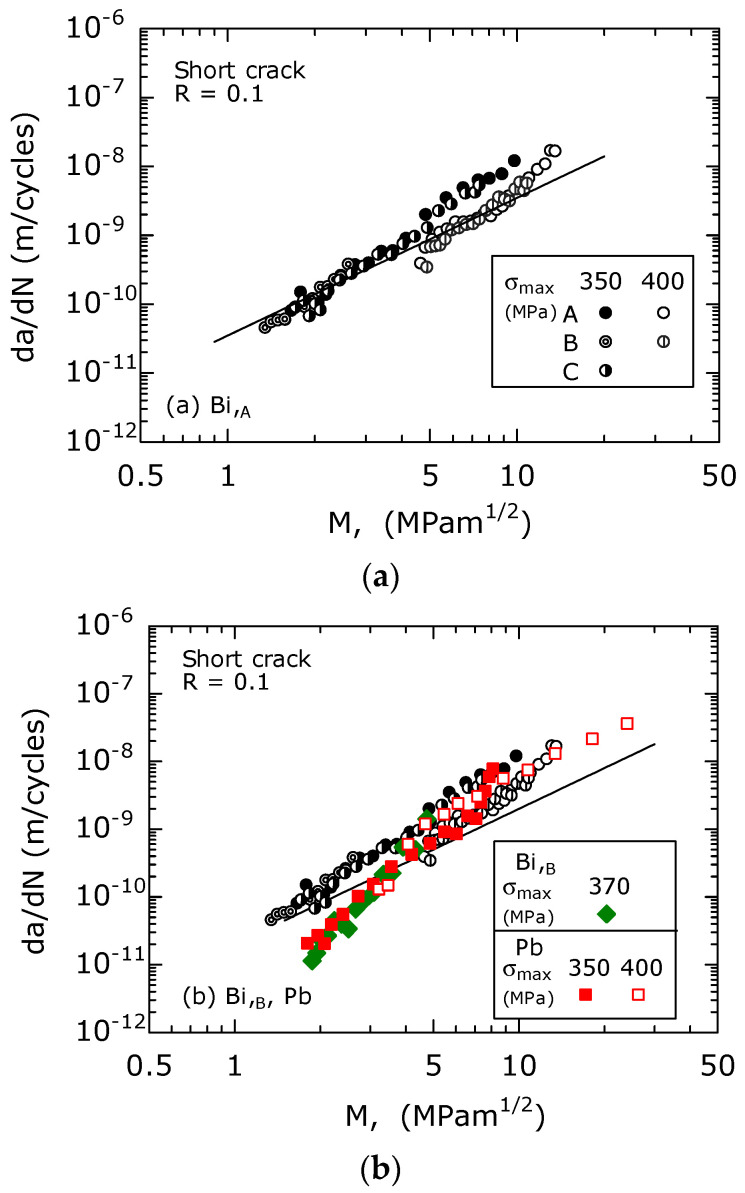
The relationship between d*a*/d*N* and *M*. (**a**) For the Bi,_A_. (**b**) For the Bi,_B_, and Pb.

**Figure 13 materials-15-07488-f013:**
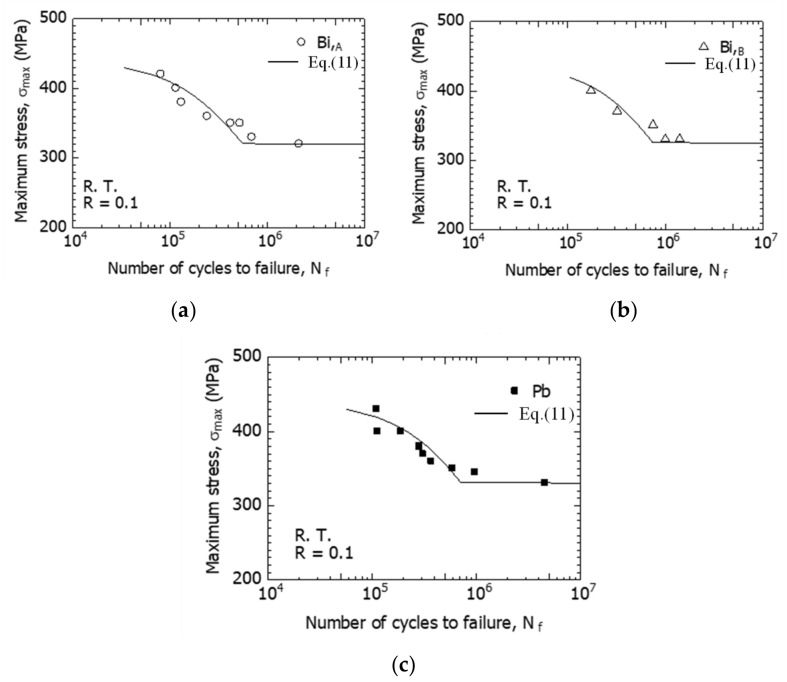
Calculation from d*a*/d*N* (*S*−*N* curves): (**a**) Bi,_A_; (**b**) Bi,_B_; and (**c**) Pb.

**Table 1 materials-15-07488-t001:** Chemical composition of the α, β, Bi and Pb phases.

Material	Phase	Phase Diameter (μm)	Cu (wt%)	Zn (wt%)	Pb (wt%)	Bi (wt%)	Sn (wt%)	Fe (wt%)
Bi,_A_ JIS C6801	α	6.9	60.86	38.87	-	0.05	0.13	0.09
	β	4.5	54.45	44.73	-	-	0.80	0.03
	Bi	-	1.32	0.53	-	97.77	0.39	-
Bi,_B_ JIS C6801	α	5.4	61.71	38.04	-	-	0.16	0.11
	β	3.9	53.73	45.52	-	0.05	0.58	0.13
	Bi	-	1.18	0.42	-	98.23	0.17	-
Pb JIS C3604	α	10.7	61.81	37.74	0.12	-	0.14	0.20
	β	3.2	53.88	45.13	0.05	-	0.89	0.06
	Pb		5.35	1.82	92.48	-	0.28	0.07

**Table 2 materials-15-07488-t002:** Mechanical properties of the materials used.

	Young’s Modulus *E* [GPa]	0.2% Stress, σ_0.2_ [MPa]	Tensile Stress, σ_B_ [MPa]	Elongation δ [%]
Bi,_A_	96	299	441	26
Bi,_B_	96	415	500	28
Pb	96	268	454	29

**Table 3 materials-15-07488-t003:** Parameter values used to calculate *M*.

Materials	Δσ_w_ [MPa]	*K*_opmax_ [MPam^1/2^]	Δ*K*_effth_ [MPam^1/2^]	*k* [m^−1^]	*r*_e_ [μm]
Bi,_A_	288	4	1.5	6000	1.11
Bi,_B_	292	4	1.5	6000	1.25
Pb	297	4	1.5	6000	1.04

## Data Availability

The data presented in this study are available upon request from the corresponding author.

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
