# Peer review of "Study on the Fatigue Crack Initiation and Growth Behavior in Bismuth- and Lead-Based Free-Cutting Brasses"

_materials, 2022, doi:10.3390/ma15217488_

Round 1

Reviewer 1 Report

The fatigue properties and mechanisms of three brass materials (Bi,A, Bi,B, and Pb materials) have been studied in depth. The obtained research results have high theoretical and application value. However, there are some issues need to be solved before acceptance as following:

1. The characterization of the different phases suggests to perform XRD analysis. Of course, it is also feasible to cite the relevant literature.

2. The font style and font size in all the Figures are as uniform as possible.

3. A number of articles recently published in this Journal on the fatigue properties and mechanisms of alloys are suggested to be refered.

4. Parts of language expressions need be improved for the whole manuscript.

Author Response

To Professor (Reviewer 1),

 We thank you for thoroughly reviewing our paper and for your valuable comments. We have responded to your comments below. In addition, we have added a few explanations to improve the paper based on your comments. We would be grateful if you could allow the publication of the paper after careful examination.

                           Dr. Kenichi Masuda

Reviewer 2 Report

The article reports the fatigue crack life and fatigue crack growth behavior of three free-cutting brasses (BiA, BiB and Pb-based materials). Both experiments and calculations were carried out to evaluate the mechanical properties of the three alloys, and results were described and summarized in a very clear and systematic way. The introduction is well written, however, some parts look more like a listing of major findings (like highlights) rather than a comprehensive summary discussing the current state of research and the perspectives in the field. Furthermore, the motivations of the study should be further emphasized and detailed as per the key assets of Bi and Pb for free-cutting brasses.

Despite a section dedicated to discussing the results, it is somehow difficult to identify clear perspectives for future research, and how these findings are relevant to the development of better free-cutting brasses (in terms of fatigue behavior and health-related aspects considering the reported harmful effect of both Bi and Pb). The Authors are thus invited to improve the discussion, and to include additional discussion correlating microstructure (phase composition) and mechanical properties, and to delve further into crack closure mechanisms (whether plastic- or roughness-induced) which could add significant depth to the discussion.

Finally, although the report is well written overall, I have serious concerns on the novelty of the work. As mentioned above it is hard to find clear perspectives, which now makes sense since according to the literature cited here, most of the pioneer work has been carried out between the 80s and early 2000.  The current findings do not seem to bring sufficient new insights on the field, except some materials properties that were not reported before according to the authors’ claims (which I encourage the Authors to verify). Given the long history of the discipline, it is striking to see that the references used are quite outdated (as evidenced above), and there is only one recent reference dating less than five years (2019). I thus have concerns on both novelty and choice of adequate references: although classical works are cited recent studies are not covered, and I believe that Authors can find better sources of information than Wikipedia (see ref.23 for instance).

A few minor comments can be additionally found below:

Line 133: what do the Authors mean by “acceptable” workability? Please elaborate

Equation 8: please revise the introducing statement “the following equation was obtained using the following”

Author Response

(The authors gave the same response as above.)

Reviewer 3 Report

The starting of abstract is some redundant- please use a specific industrial back ground rather then stressing there are only few study. Even there may are few study, they can studied this !

Some results in brief in abstract are require !

The first paragraph in introduction do no make a proper flow! Please revise it

Line 43 is bringing some weak formulation ! which only few study were discussed ?

Overall the introduction requires a better structure and presentation. IT is repeated that there only few study evaluate this topic however your research is not very systematic !

You have indicated different phase for Figure 1 however these comes from optical microscopy which may are not right elements !They have to be endorsed by literature data or EDS !

In methods and material section were mixed with results ! please revise as it is not a proper research structure !

Not clear which were the imposed condition in order to obtain the data from Table 2. For example loading condition, speed, cross head and also geometry of samples used !

Not clear if the specimens used for fatigue, figure 3 corresponds to any standard ? if so please specify it

The fatigue fracture surfaces were observed using SEM” however the image is rather from optical microscope ! (figure 4)

You said about grain boundary facets in Figure 6c but this is not actually the right thing ! also about stripped pattern is not correct !

I am curious to see the Figure 10 a without red line about crack – I am suspecting this is not correct what you indicate !

“Mr. Shinya Yamamoto” not clear what was contribution to this person, but I feel he/she should appear on the authors list !

I expected to see some recent references – this denotes that the authors have a weak evaluation of literature review

Author Response

(The authors gave the same response as above.)

Round 2

Reviewer 3 Report

.